# Structure, Morphology, Heat Capacity, and Electrical Transport Properties of Ti_3_(Al,Si)C_2_ Materials

**DOI:** 10.3390/ma14123222

**Published:** 2021-06-11

**Authors:** Kamil Goc, Janusz Przewoźnik, Katarzyna Witulska, Leszek Chlubny, Waldemar Tokarz, Tomasz Strączek, Jan Marek Michalik, Jakub Jurczyk, Ivo Utke, Jerzy Lis, Czesław Kapusta

**Affiliations:** 1Faculty of Physics and Applied Computer Science, AGH University of Science and Technology, Al. Mickiewicza 30, 30-059 Kraków, Poland; januszp@agh.edu.pl (J.P.); tokarz@agh.edu.pl (W.T.); Tomasz.Straczek@agh.edu.pl (T.S.); jmichali@agh.edu.pl (J.M.M.); Jakub.Jurczyk@agh.edu.pl (J.J.); kapusta@agh.edu.pl (C.K.); 2Faculty of Materials Science and Ceramics, AGH University of Science and Technology, Al. Mickiewicza 30, 30-059 Kraków, Poland; chabior@agh.edu.pl (K.W.); leszek@agh.edu.pl (L.C.); lis@agh.edu.pl (J.L.); 3Laboratory for Mechanics of Materials and Nanostructures, Empa-Swiss Federal Laboratories for Materials Science and Technology, Feuerwerkerstrasse 39, 3602 Thun, Switzerland; ivo.utke@empa.ch

**Keywords:** MAX phases, hot pressing synthesis, heat capacity, electrical properties, magnetoresistance, DOS calculations

## Abstract

A study of Ti_3_Al_1−*x*_Si*_x_*C_2_ (*x* = 0 to *x* = 1) MAX-phase alloys is reported. The materials were obtained from mixtures of Ti_3_AlC_2_ and Ti_3_SiC_2_ powders with hot pressing sintering technique. They were characterised with X-ray diffraction, heat capacity, electrical resistivity, and magnetoresistance measurements. The results show a good quality crystal structure and metallic properties with high residual resistivity. The resistivity weakly varies with Si doping and shows a small, positive magnetoresistance effect. The magnetoresistance exhibits a quadratic dependence on the magnetic field, which indicates a dominant contribution from open electronic orbits. The Debye temperatures and Sommerfeld coefficient values derived from specific heat data show slight variations with Si content, with decreasing tendency for the former and an increase for the latter. Experimental results were supported by band structure calculations whose results are consistent with the experiment concerning specific heat, resistivity, and magnetoresistance measurements. In particular, they reveal that of the s-electrons at the Fermi level, those of Al and Si have prevailing density of states and, thus predominantly contribute to the metallic conductivity. This also shows that the high residual resistivity of the materials studied is an intrinsic effect, not due to defects of the crystal structure.

## 1. Introduction

MAX phases belong to the ternary and quaternary group of materials (carbides and nitrides). They can be specified by the main formula M*_n_*_+1_AX*_n_*, where M is an early transition metal, A stands for an A-group element (Al, Si), and X is carbon and/or nitrogen. These compounds crystallize in the hexagonal system P6_3/mmc_ and they possess a nanolaminated structure with coexisting different types of bonds between elements. As a result of their heterodesmic structure, some properties of MAX phases, like high thermal and electrical conductivity, as well as a ductile nature, derive from metals. On the other hand, they can behave as typical ceramics, e.g., they have high stiffness, low thermal expansion coefficient, and perfect thermal and chemical resistance, even at high temperatures. Owing to this combination of untypical properties, MAX phases are located between ceramic and metallic materials [1,2,3,4]. Their properties make them useful in applications like conductive coatings, heat proof materials, thermal shock resistant composites, and neutron irradiation resistive systems [1,5,6].

The unit cell of these materials is built of M_6_X octahedrons, which are located between A-atom layers and the atoms of carbon/nitrogen occupying the centers of octahedrons. Referring to the main formula M*_n_*_+1_AX*_n_*, where *n* = 1, 2, 3, …, different types of MAX phases are labelled as 211, 312 and 413, respectively. It means that there are two M-layers separating the A-layers in the 211 group, whereas in 312s there are three and in 413s, there are four M-layers. According to the literature, ordered structures with *n* = 4, 5, 6 and even 7, creating 514, 615 and 716 MAX phases have also been discovered. A number of papers report on several methods, which can be used to obtain them as bulk materials. They include hot isostatic pressing [7,8,9], microwave sintering [10], arc melting [11,12], annealing [13] spark plasma sintering [14,15,16,17], and self-propagating high-temperature synthesis (SHS) [18,19,20,21,22,23,24], usually followed by hot pressing sintering [18,22,25]. To obtain them in the form of thin films, physical or chemical vapour-deposition (PVD or CVD) can be used [2,12,26].

Previous studies have shown that in these compounds many substitutions on M, A, and X position are possible. Doping elements usually come from the neighbourhood in the periodic table, due to their close atomic radii and electronic similarity. About 100 sets of compounds of solid solutions, like Cr_2_(Al_1−*x*_Si*_x_*)C, Ti_3_(Al_x_Sn_1−*x*_)C_2_, Ti_2_Al(C_0.5_N_0.5_), Ti_3_(Al_0.75_Si_0.25_)C_2_ and others have been fabricated and examined so far. Moreover, solid solutions of MAX phases often present more attractive physical, chemical, and mechanical properties compared to the end-member materials [7,27,28,29,30,31,32].

At present, the most investigated pure MAX phases, belonging to the 312 family (Figure 1), are Ti_3_AlC_2_ and Ti_3_SiC_2_. Consequently, the next step of MAX phases development was to synthesize and investigate solid solution-materials in the Ti-Al-Si-C system. Usually, the Ti_3_(Al_1−*x*_Si*_x_*)C_2_ compounds are synthesized from pure elemental powders with different stoichiometry *x* [31]. Following our recent work on Ti_3_AlC_2_ material [19], in this paper, we present results on the synthesis and characterization of three different Ti_3_(Al_1−*x*_Si*_x_*)C_2_ materials with stoichiometry *x* = 1/3, 1/2 and 2/3.

The mixed compounds prepared within the present work have been produced by hot pressing synthesis from the Ti_3_AlC_2_ and Ti_3_SiC_2_ powders. Hot-Pressing (HP) is a method in which high temperature and uniaxial pressing are applied simultaneously to the powder in order to obtain dense material. The two starting materials have been synthesized by SHS from the appropriate mixture of the TiAl, Ti, Si, and C powder components. For the dense materials of MAX phase compounds obtained, the X-ray diffraction (XRD), scanning electron microscopy (SEM), heat capacity, and electrical resistivity/magnetoresistance measurements have been carried out. The experimental characterization has been supported with calculations of the local densities of electronic states.

From the XRD patterns, the crystal structure parameters of the compounds have been determined. SEM and energy dispersive X-ray spectroscopy (EDS) analysis have been used for the evaluation of the microstructure and the elements concentration. A specific heat study was carried out to obtain information on the contributions of the electronic and the lattice subsystems. Measurements of the electrical resistance and magnetoresistance in the temperature range 4–300 K and at the magnetic field up to 9 Tesla were done to get the information on the type of the electrical conductivity and the mobility of the carriers. The results of measurements are discussed with respect to the local density of states calculations. The relation to the thermal and electrical properties of the materials studied is analysed and discussed.

## 2. Experimental

For the synthesis of the Ti_3_(Al_1−*x*_Si*_x_*)C_2_, mixtures of SHS derived powders of Ti_3_AlC_2_ and Ti_3_SiC_2_ were used, according to the intended stoichiometry, as follows:^2^/_3_ Ti_3_AlC_2_ + ^1^/_3_ Ti_3_SiC_2_ → Ti_3_Al_2/3_Si_1/3_C_2_(1)
^1^/_2_ Ti_3_AlC_2_ + ^1^/_2_ Ti_3_SiC_2_ → Ti_3_Al_1/2_Si_1/2_C_2_(2)
^1^/_3_ Ti_3_AlC_2_ + ^2^/_3_ Ti_3_SiC_2_ → Ti_3_Al_1/3_Si_2/3_C_2_(3)

All the mixtures (6 g each) were homogenized in a ball-mill (Gabrielli, Calenzano, Italy) with tungsten carbide balls for 24 h. After that, the homogenized mixtures were placed in a 1-inch diameter graphite mould and subjected to the hot-pressing process. Samples stayed in the hot pressing (HP) apparatus for 1 h at maximum temperature and pressure, 1300 °C and 25 MPa, respectively. After completion of the reactive pressing process, the graphite residues were removed from the samples surface.

In order to determine the phase composition of both, starting powders and hot-pressed bulk materials, the XRD measurements were carried out at room temperature in Bragg-Brentano geometry using Cu K_α_ radiation. The Siemens D5000 diffractometer (Siemens AG, Berlin, Germany) equipped with a diffracted beam graphite monochromator (using zero background sample holder) was used for the powders and the Panalytical Empyrean diffractometer was used for the bulk samples. Mass contributions of the respective phases were derived according to Rietveld analysis [34]. Electrical resistivity, magnetoresistance, and heat capacity measurements were carried out, respectively, with the AC transport and heat capacity options of a Physical Property Measurement System, PPMS (Quantum Design, San Diego, CA, USA) apparatus. The samples for the electrical resistivity and magnetoresistance measurements were cut from the pellets with a diamond saw to obtain parallelepipeds (1 mm × 1 mm × 10 mm), with their longest edge parallel to the pellet plane. The density of states calculations were done with Wien2K software (version 14) [35], which is suitable for periodic structures as it uses periodic boundary conditions. SEM secondary electron images were taken on a JEOL 5900LV microscope (JEOL Ltd, Tokyo, Japan) and the EDS analysis was performed using the Noran type 6 equipment (Thermo Fisher Scientific, Waltham, MA, USA).

## 3. Characterization

### 3.1. Characterization of the Crystallographic Structure

The Rietveld analysis of the XRD powder patterns for the starting Ti_3_AlC_2_ and Ti_3_SiC_2_ materials revealed some admixtures of impurity phases; see Figure 2. In the case of Ti_3_AlC_2_ material, the 12.3 ± 0.7% mass contribution of a secondary TiC cubic phase with unit cell parameter *a* = 4.3219 ± 0.0002 Å was found. The analysis of the XRD powder pattern of Ti_3_SiC_2_ material revealed 26.5 ± 1.2% contribution of the TiC phase and 11.5 ± 1.0% contribution of the TiSi_2_ phase with orthorhombic (space group *F ddd*) unit cell, with *a* = 8.260 ± 0.002 Å, *b*= 4.800 ± 0.001 Å and *c* = 8.549 ± 0.003 Å. 

The XRD patterns, along with the Rietveld refined curves for the HP synthesized Ti_3_(Al_1−*x*_Si*_x_*)C_2_ bulk samples, are presented in Figure 3. It has to be noted that in all patterns, besides the MAX phase, the impurity TiC and Al_2_O_3_ (corundum) with trigonal (space group *R* $\bar{3}$*c*) phases were also detected. 

The calculated mass contributions of the phases are collected in Table 1 and the structural data corresponding to the majority Ti_3_(Al_1−*x*_Si*_x_*)C_2_ hexagonal phases are listed in Table 2. Materials contain about 90% of MAX phase and it can be noticed that this is slightly higher than for the starting materials, indicating that during the hot pressing process, the TiC and TiSi_2_ impurities convert into the MAX phase. A similar effect was observed in our previous study [19]. Narrow lines of XRD patterns reveal their good crystallinity. Spurious phases present in the amount less than 10% in the samples of solid solution materials, which are mainly TiC, do not influence significantly the stoichiometry and properties of the main MAX phase.

The dependencies of the hexagonal unit cell volume V and the lattice parameters *a*, *c* on the Si content *x* determined for all the Ti_3_(Al_1−*x*_Si*_x_*)C_2_ samples are shown in Figure 4. In Figure 4a, the deviation of the unit cell volume from Vegard’s law (linear dependence with *x*) is also marked. It is worth noting that the largest difference in volume of about 1.3 Å^3^ is visible for the *x* = 0.5 sample. 

A similar deviation from Vegard’s law is seen for the lattice parameter *c*(*x*) shown in Figure 4b. In contrast, the opposite curvature, a more linear character, and a much weaker dependence of the lattice parameter *a*(*x*) was found. It is worth noting that the relative contraction of the c parameter (Δc/c) between the end compounds Ti_3_AlC_2_ and Ti_3_SiC_2_ is more than 20-times larger than the corresponding contraction (Δa/a) for the a parameter. Such anisotropic contraction can be expected for the layered type structure of these compounds when substitution of Al atoms forming separate layers along the *c* direction of the hexagonal unit cell by much smaller Si atoms takes place.

### 3.2. Microstructure Characterization

In order to study the morphology of the materials, SEM and EDS measurements were carried out. A representative example image of the surface of the sample is presented in Figure 5 for the Ti_3_Al_1/3_Si_2/3_C_2_ material. It shows a flat structure decorated with small pores and ruptures typical for saw-cut materials synthetized with the hot-pressing method. The higher magnification (right panel) reveals the shape and size of the MAX phase grains, which is about 1 to 10 μm, similar to that reported by us in [19] for the border compound, Ti_3_AlC_2_.

EDS spectra presented in Figure 6 show the intensities of the fluorescence lines of the constituent elements, which represent their relative content in the material. As can be expected, after normalization of the spectra to the maximum peak (Ti line), the silicon and alumina intensities grow/decrease with *x*. The example of a linear scan with EDS, shown in the insets, reveals an even distribution of elements, indicating good homogeneity of the material.

### 3.3. Heat Capacity

Figure 7 shows the temperature dependence of the specific heat ($C_{p}$) for Ti_3_AlC_2_ and Ti_3_Al_1−*x*_Si*_x_*C_2_ bulk compounds. One should note their close resemblance, but also clear differences at some temperature ranges. At high temperatures, the experimental points for all the compounds studied practically overlap, but they start to depart below 100 K (see top inset in Figure 7). At the low temperature range (*T* < 10K), they show quite distinct differences (see bottom inset in Figure 7). The linear temperature dependence of the $C_{p}/T$ versus $T^{2}$ at the low temperature region corresponds to a Debye-like character of the lattice contribution. The coordinates of the extrapolated y-avis crossing points provide Sommerfeld coefficient values of the electronic contribution, revealing a metallic character of the MAX phase compounds studied. 

The $C_{p}$ vs. T dependence of the total specific heat of the nonmagnetic Ti_3_(Si,Al)C_2_ compounds can be approximated by phonon and electronic ($C_{el}=\gamma T$) contributions using the following expression [36,37]:(4)$$C_{p}=C_{ph}+C_{el}=\frac{R}{1-\alpha T}\left[9{\left(\frac{T}{\Theta_{D}}\right)}^{3}\overset{\Theta_{D}/T}{\underset{0}{\int }}\frac{x^{4}e^{x}}{{\left(e^{x}-1\right)}^{2}}\mathrm{dx}+\underset{i}{\sum }\frac{m_{i}{\left(\;\frac{\Theta_{E_{i}}}{T}\right)}^{2}e^{\Theta_{E_{i}}/T}}{{\left(e^{\Theta_{E_{i}}/T}-1\right)}^{2}}\right]+\gamma T,$$
where $\Theta_{D}$ is the Debye temperature, $\Theta_{E_{i}}$ are Einstein temperatures, $m_{i}$ are corresponding multiplicities for each individual optical branch, α stands for an anharmonic coefficient, γ is an electronic specific heat (Sommerfeld) coefficient, and R is the gas constant. It is assumed that there are three acoustic modes in the compounds (described by the first term in square brackets) and 15 optical modes (described by the second one). In order to facilitate analysis, the summation over 15 independent optical branches was grouped into four branches of 3 or 6-fold multiplicity (this grouping was optimized by allowing for refinement of $m_{i}$ parameters during preliminary fitting of the formula and then rounding and fixing their values to the nearest natural number in the final fit). 

To determine the approximated Debye temperatures ($\Theta_{D}^{l}$) and Sommerfeld (γ) coefficients conveniently, we can approximate Equation (4) at low T by the sum of Debye $T^{3}$ law and γT linear term and fit linear parts of the low-temperature $C_{p}/T$ vs. $T^{2}$ plots of the same data (shown in the bottom inset in Figure 7. The values of the $\Theta_{D}^{l}$ and γ parameters determined from them are listed in Table 3. One should note the very close values of $\Theta_{D}^{l}$ and γ parameters for the Ti_3_AlC_2_ and Ti_3_Al_1/2_Si_1/2_C_2_ compounds. Lower values of $\Theta_{D}^{l}$ for the Ti_3_Al_2/3_Si_1/3_C_2_ and Ti_3_Al_1/3_Si_2/3_C_2_ compounds indicate a softer lattice in them. The value of γ, which is noticeably larger for Ti_3_Al_1/3_Si_2/3_C_2_ than for the other compounds, correlates well with a higher DOS at Fermi level obtained for this compound (see Section 3.5). 

From the fitting of Equation (4) to the whole $C_{p}$ vs. T dependences from Figure 7 (in the refinement the Sommerfeld coefficients γ were fixed at corresponding values from Table 3), the Debye ($\Theta_{D}$), Einstein ($\Theta_{E_{i}}$) temperature parameters and anharmonic coefficients (α) were also obtained for the Ti_3_AlC_2_ and Ti_3_Al_1/2_Si_1/2_C_2_ compounds; they are provided in Table 4. One should note clearly lower values of the Debye temperatures ($\Theta_{D}$) from Table 4 (full temperature range fit) than from Table 3 (low temperature range fit). The latter can be regarded as representative, since the full temperature range fits a much larger error of the model, which involves many Einstein terms. 

Figure 8 presents the variation of the experimental value of γ with Si content, together with theoretical values obtained from calculations of the electronic density of states, presented in details in the subsequent paragraph. 

The experimental values are slightly higher than the theoretical ones, but the tendency with increasing Si doping is similar. It is worth noting that the high Si doping corresponds to a significantly increased γ. 

### 3.4. Electrical Resistivity and Magnetoresistance

Measurements of the electrical resistivity were conducted with the four-point-probe method of the ACT (Alternating Current Transport) option of the QD PPMS (Quantum Design Physical Property Measurement System) apparatus in the temperature range from 4 K to 300 K. The results are presented in Figure 9. 

They show a linear dependence down to the temperature of 100 K, which reveals the metallic character of the materials. The residual resistivity, however, is relatively high and corresponds to a small Residual Resistivity Ratio (RRR), i.e., R(293K)/R(4K), of 3. The low RRR is usually related to a substantial contribution of the scattering of electrical carriers at the crystal defects, comparable to that on phonons, dominating at high temperatures. Low RRR values have also been observed in our previous studies, [19] and by Finkel et al. [38], where for Ti_3_AlC_2_, a value of 1.95 was reported. 

In order to shed more light on the electrical transport in the materials, measurements of the electrical resistivity have also been carried out at the applied magnetic field, to determine the magnetoresistance [39], i.e., the (R_H_-R_0_)/R_0_ values at different temperatures. The results obtained are shown in Figure 10.

The magnetoresistance at 300 K is much smaller than at 4 K for the materials studied, as should be expected. The field dependencies of the magnetoresistance are found to fit well to a quadratic function, which indicates that the magnetoresistance effect in the MAX phase materials studied is dominated by contribution from open orbits. This is confirmed by the shape of the Fermi surface (Figure 11) derived from calculations [40] of the electronic structure for the end member compounds of the series, described in the paragraph below. It shows that the Fermi surface largely intersects the first Brillouin zone boundary along the [1] direction for Ti_3_AlC_2_ and, for Ti_3_SiC_2_, also in the directions within the perpendicular hexagonal plane. This brings about a large contribution of open orbits to the magnetoresistance, resulting in its quadratic dependence on the magnetic field. 

### 3.5. Density of States Calculations

The calculations were done in the full potential WIEN 2k code [35] based on the density functional theory (DFT) [41,42] and the generalized gradient approximation (GGA) [43]. Table 5 presents the Ti_3_AlC_2_ structural data used, i.e., the lattice constants and Wyckoff position of atoms in the elementary cell. These parameters were obtained after minimization of total energy as a function of volumes and *c*/*a* with starting values taken from [20]. All free parameters in site positions were adjusted according to force minimization. For calculations of P63/mmc structure (ITC 194) with supercell, a 2 × 2 × 1 and parameters listed in Table 5 was used. For the Ti_3_Al_1/2_Si_1/2_C_2_ sample, a 2 × 2 × 1 superstructure within ICT 194 was adopted with shifting by <00¼> vector and respective replacement of the Si and Al sites. We obtained the appropriate ICT 187 structure, which was used for the calculations. 

The reduced unit cell volume and the lattice constants calculated show a decrease with increasing Si content, as did the experimental values. However, the *a* parameter shows a zig-zag dependence on x, with an excess value for the Ti_3_Al_1/4_Si_1/4_C_2_ sample. This feature is also visible in the unit cell volume dependence. To explain the possible origin of this effect, we also did the calculations for the Ti_3_Al_1/2_Si_1/2_C_2_ in the structure 187 with Al and Si in separate layers and got *a* = 3.0870 Å; however, but if we mix Si and Al within layers into the 2 × 2 × 1 superstructure, this parameter is lower and amounts to 3.0517 Å. Thus, if the Al/Si occupation in the Ti_3_Al_1/4_Si_3/4_C_2_ was fully random, which is possibly the case, we would probably get a lower *a* parameter value, so our supercell model may not precisely describe the real material. 

The calculated electronic structures of all the compounds (*x* = 0, 1/4, 1/2, 3/4 and 1) show several bands crossing the Fermi level, which reveals their metallic character. The examples of the structure are shown for the end members of the series in Figure 11. The Fermi surface for an exemplary band of Ti_3_AlC_2_ was already presented in [19] and in the present paper complete Fermi surfaces for both compounds are shown, with the individual bands marked as a, b, c …

Usually the electrical properties of metallic-like materials relate the carrier density to the density of states at the Fermi energy, DOS(E_F_). Thus, a considerable density of states at E_F_ obtained for Ti_3_AlC_2_, much larger than that for aluminium metal, should result in a very good conductivity, which is obviously not the case. This is due to the fact that the most effective current carriers are those from s-band, which is the most delocalised, while the d-band electrons act as scatterers, decreasing the conductivity. For titanium metal, this effect is very dramatic and leads to forty times lower conductivity than that of copper, where the 3d band is filled and has a negligible DOS at the Fermi level. This is very different from the situation in specific heat, where all the electrons at the Fermi level take part and the Sommerfeld coefficient of the electronic part is much higher for titanium than for copper, as mentioned above. A closer inspection of Figure 12 shows that the dominant densities of states at E_F_ for Ti-1 and Ti-2 sites are predominantly of d-type and are nearly two orders of magnitude higher than those of the p- or s-band. In contrast, for the Al/Si site, the s-band DOS(E_F_) is an order of magnitude higher than for both sites of Ti or C. It is also comparable to that in Al metal. Thus, the propagation of the electrical carriers provided by the Al/Si sites is obstructed by the low s-like DOS(E_F_) at the adjacent Ti and C sites, which results in high resistivity.

The magnetoresistance effect in a metal is approximately inversely proportional to the relaxation time of the electrical carriers, i.e., directly proportional to their mobility. From the above discussion, a mean free path of the order of interatomic distances can be anticipated, corresponding to a short relaxation time. This is consistent with the large residual resistivities and very low values of the magnetoresistance. In view of good crystallinity of the materials, as revealed in the X-ray diffraction study, the electron carrier scattering at the lattice defects can be excluded as the dominant mechanism. Considering quasi two-dimensionality of the materials, the mechanism related to the Anderson localization should also be taken into account [44,45,46]. However, its verification is beyond the scope of this work. The picture of the electronic properties drawn from the above presented and discussed results is consistent with a relatively small electronic contribution to the specific heat, indicating a limited population of delocalised electrons at the Fermi level taking part in the electrical and thermal phenomena.

## 4. Conclusions

The study carried out on the Ti_3_(Al,Si)C_2_ materials synthesized with hot pressing from Ti_3_AlC_2_ and Ti_3_SiC_2_ powders obtained with self-propagating high-temperature synthesis revealed their good crystallinity, as deduced from narrow lines of their X-ray diffraction patterns.

Measurements of specific heat delivered the Debye temperatures and Sommerfeld coefficient values, which show slight variations with Si doping with decreasing tendency for the Debye temperatures and an increase for the Sommerfeld coefficient. A small residual resistivity ratio of 3 was obtained from the measurements. The magnetoresistance at 4 K was an order of magnitude larger than at 300 K and was found to exhibit a quadratic field dependence. This reveals a dominant contribution coming from open electronic orbits and is consistent with the calculated shape of the Fermi surface.

From a detailed look at the crystallographic and electronic structure, specific heat, electrical resistivity, and magnetoresistance of the materials, a novel result of the present contribution could be drawn. It relies on the explanation of the relatively high resistivity of MAX phases with the disparity of the local densities of states at the Fermi level derived for individual sites and orbitals of constituent elements. This concerns the densities of s-electrons, which contribute the most to the electrical conductivity and are provided there mainly by aluminium and silicon. Neighbouring titanium atoms with very low s-electron densities, but very high d-electron densities act as scatterers for the electrical carriers. This draws a consistent picture of the electronic and thermal properties of the MAX phase materials studied and the intrinsic character of their high residual resistivity.

## Figures and Tables

**Figure 1 materials-14-03222-f001:**
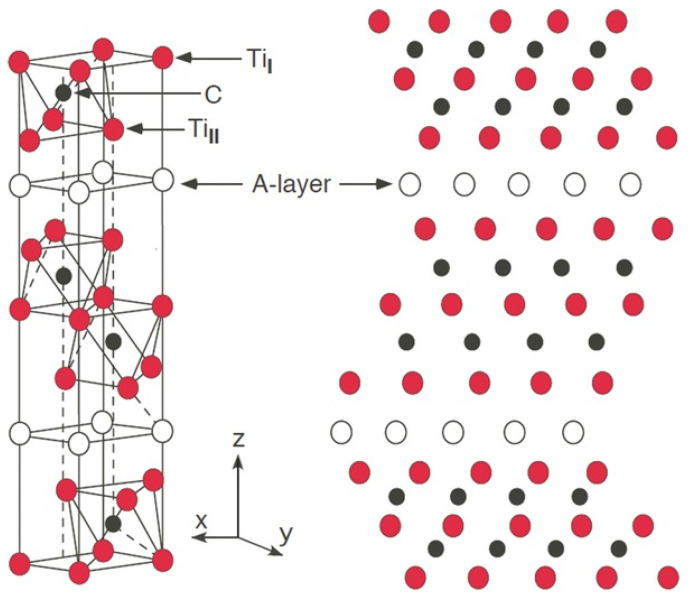
Crystal structure of a 312 MAX phase. Ti atoms occupy two different sites, Ti_I_ and Ti_II_. A layer of A-group atoms appears after each third Ti layer [33].

**Figure 2 materials-14-03222-f002:**
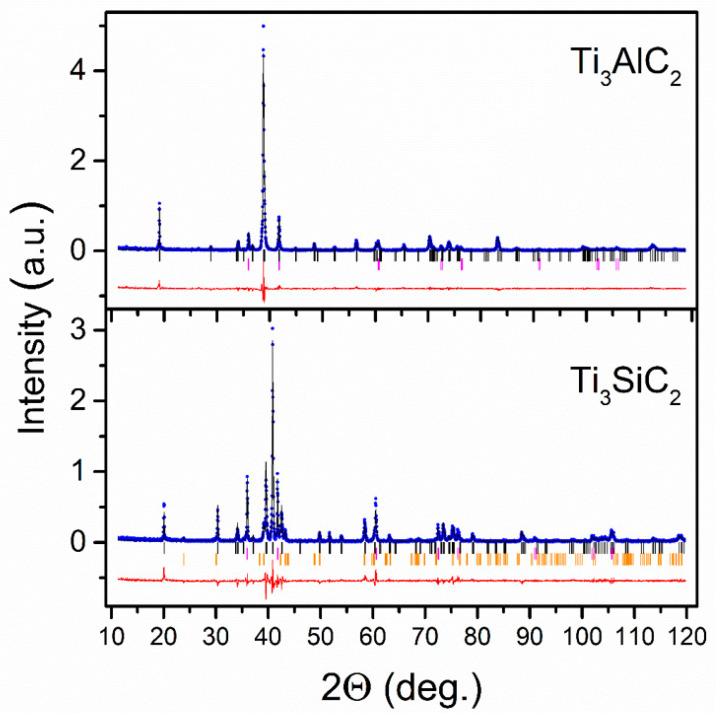
XRD patterns (blue dots) and the best-fit Rietveld refinement (solid black lines) of the Ti_3_AlC_2_ and Ti_3_SiC_2_ starting materials. The rows of vertical bars denote the Bragg peak positions for the corresponding MAX phase (black), TiC (magenta), and TiSi_2_ (orange). The solid red line represents the difference between the experimental and calculated patterns.

**Figure 3 materials-14-03222-f003:**
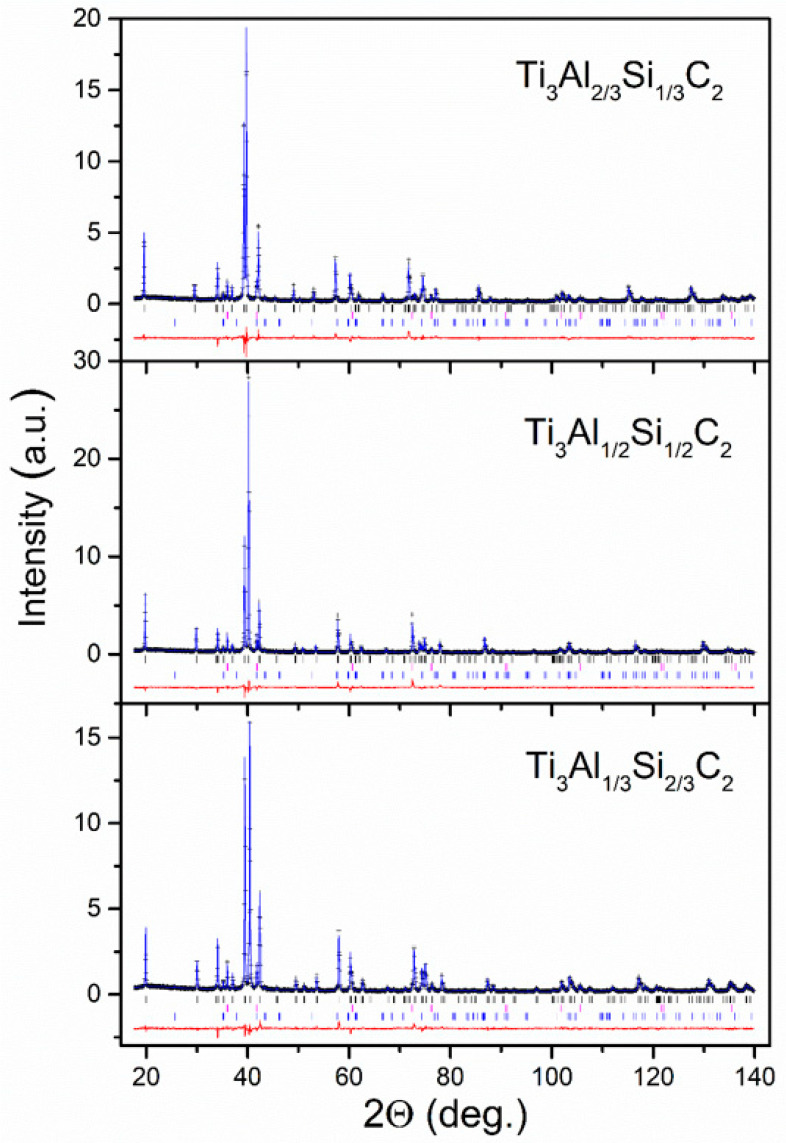
XRD patterns (black crosses) and the best-fit Rietveld refinement (solid blue lines) of the Ti_3_Al_1−*x*_Si*_x_*C_2_ bulk sample (pellet). The rows of vertical bars below denote the Bragg peak positions for corresponding MAX phase (black), TiC (magenta), and Al_2_O_3_ (blue) phases. The solid red lines represent the difference between the experimental and calculated patterns.

**Figure 4 materials-14-03222-f004:**
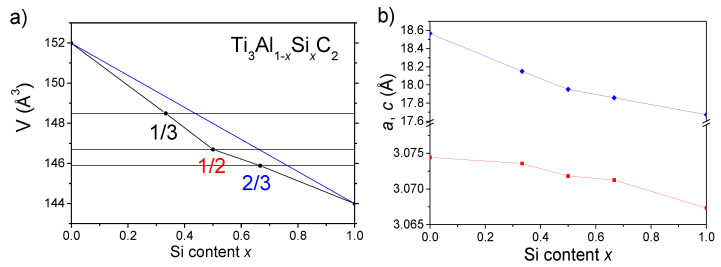
(**a**) Dependence of the unit cell volume V on the Si content x of the Ti_3_(Al_1−*x*_Si*_x_*)C_2_ samples (black dots). The blue solid line represents an assumed linear V(x) dependence. (**b**) Dependence of the unit cell parameters *a* (red squares) and *c* (blue diamonds) on the Si content *x* of the Ti_3_(Al_1−*x*_Si*_x_*)C_2_ samples. Errors are comprised in symbol size.

**Figure 5 materials-14-03222-f005:**
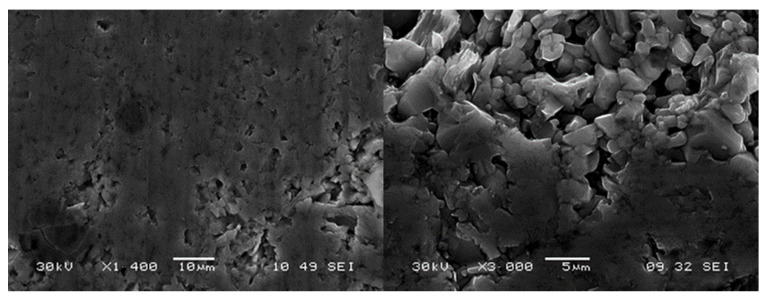
SEM images of the Ti_3_Al_1/3_Si_2/3_C_2_ material surface.

**Figure 6 materials-14-03222-f006:**
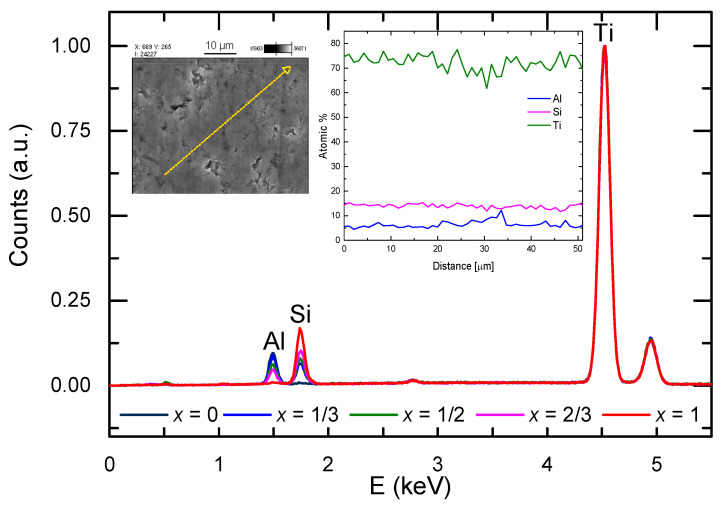
EDX spectra of the Ti_3_(Al_1−x_Si_x_)C_2_ samples. The insets show the SEM image of the surface of Ti_3_(Al_1/3_Si_2/3_)C_2_ (the arrow marks the line of the EDS scan) and the plot of concentrations of Ti, Al, and Si along this line.

**Figure 7 materials-14-03222-f007:**
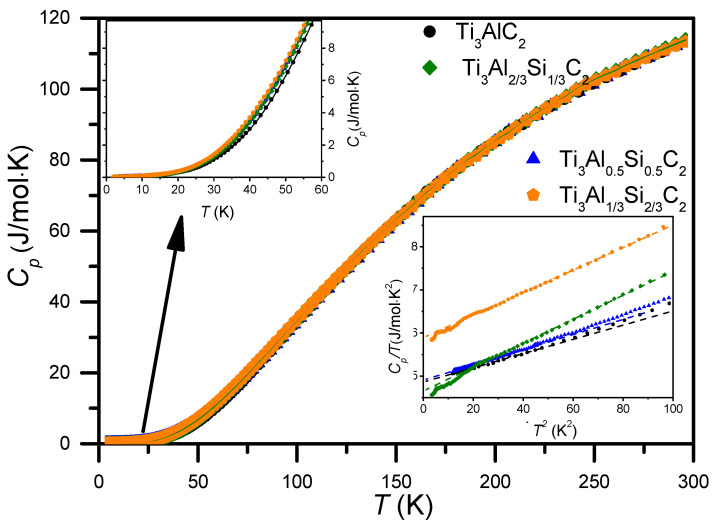
Temperature dependencies of the specific heat ($C_{p}$) of Ti_3_AlC_2_, and Ti_3_Al_1−*x*_Si*_x_*C_2_ compounds. Fits (solid lines) represent Equation (1). The top inset shows the expanded low *T* range. The bottom inset shows the low *T* range as $C_{p}/T$ vs. $T^{2}$ plots. Linear fits to the data in the inset (dashed lines) were used to estimate the Debye temperature ($\Theta_{D}$) and the Sommerfeld coefficient (γ) of the compounds.

**Figure 8 materials-14-03222-f008:**
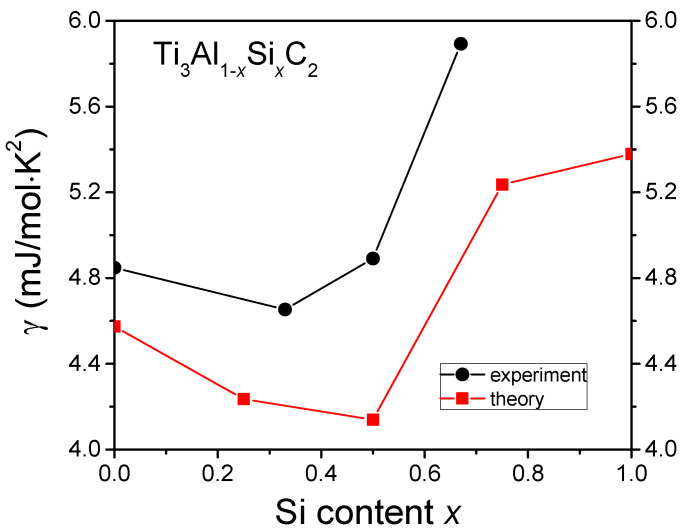
Dependence of the γ values on the Si content obtained from the experiment and calculations.

**Figure 9 materials-14-03222-f009:**
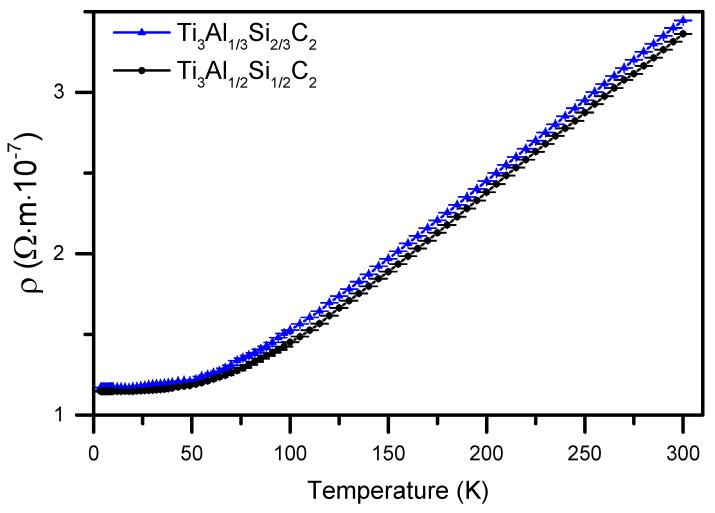
Electrical resistivity of Ti_3_(Al,Si)C_2_ materials as a function of temperature. Experimental points with error bars are marked.

**Figure 10 materials-14-03222-f010:**
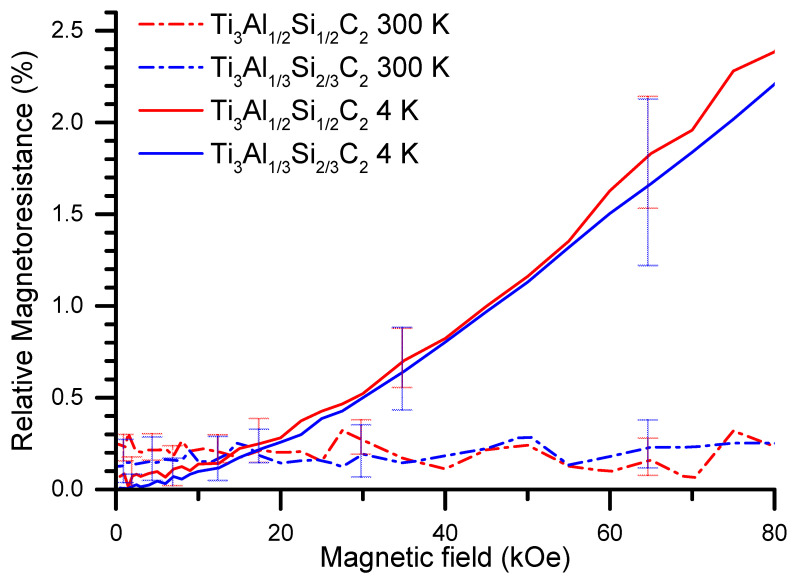
Magnetoresistance vs. magnetic field strength of the MAX phase materials at 300 K and 4 K.

**Figure 11 materials-14-03222-f011:**
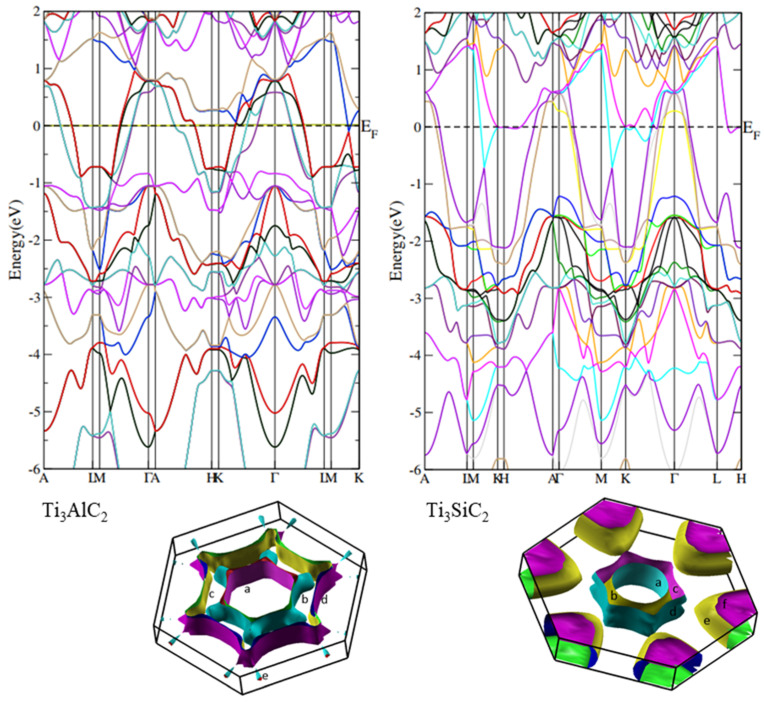
Band structures and Fermi surfaces of Ti_3_AlC_2_ (left) and Ti_3_SiC_2_ (right) for different bands. Vertical direction corresponds to the <001> axis in the reciprocal lattice.

**Figure 12 materials-14-03222-f012:**
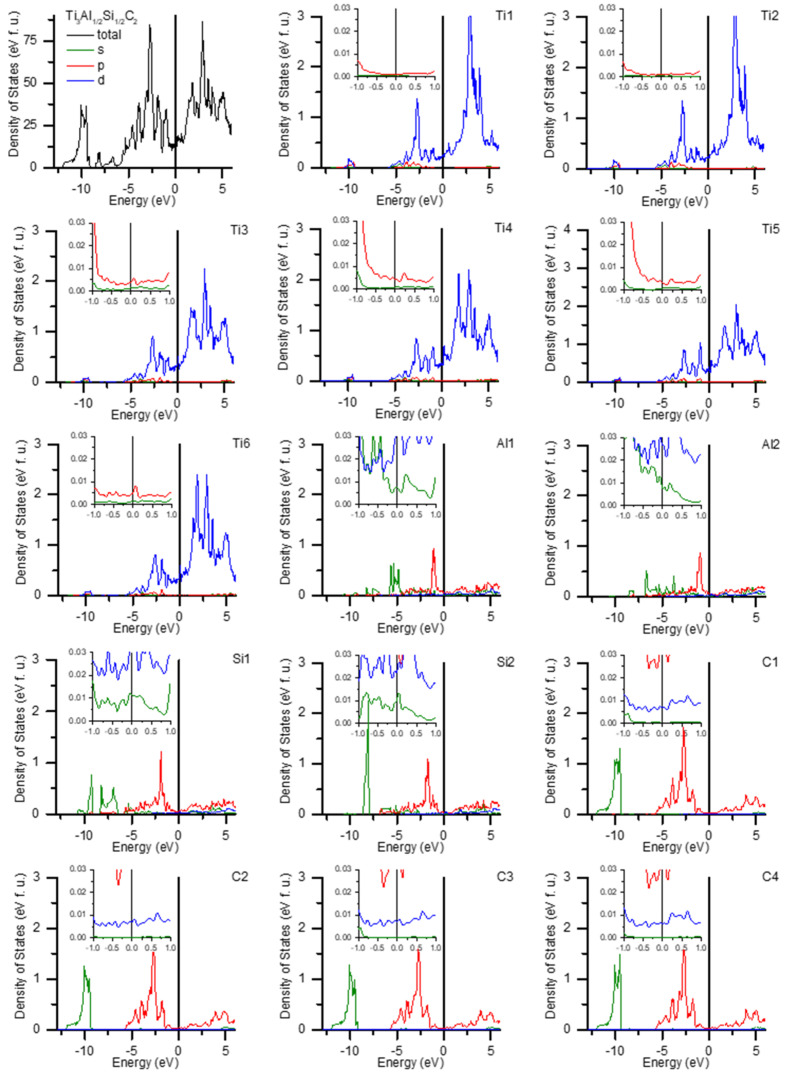
Calculated density of states of the Ti_3_Al_1/2_Si_1/2_C_2_ compound. Densities of states for s (green), p (red), and d (blue) sub-bands and the total density of states (black) are presented.

**Table 1 materials-14-03222-t001:** Mass contributions (%) of the phases obtained from the analysis of X-ray diffraction patterns of Ti_3_(Al_1−*x*_Si*_x_*)C_2_ bulk samples with *x* = 0, 1/3, 1/2, 2/3, where Ti_3_(Al_1−*x*_Si*_x_*)C_2_ is the majority hexagonal (space group P 6_3/mmc_) phase and TiC cubic (space group F m$\bar{3}$m) and Al_2_O_3_ trigonal (space group R $\bar{3}$c) are secondary phases.

*x* (Si at.%)	Ti_3_(Al_1−*x*_Si*_x_*)C_2_	TiC	Al_2_O_3_	TiSi_2_
0	88(3)	12.3(7)	0	-
1/3	89(2)	6(1)	5(1)	0
1/2	87(2)	8(1)	3(1)	0
2/3	91(2)	6(2)	3(1)	0
1	62(3)	27(2)	-	11(1)

**Table 2 materials-14-03222-t002:** Structural data of the dominant Ti_3_(Al_1−*x*_Si*_x_*)C_2_ hexagonal phase (space group P 6_3_/mmc) obtained from the Rietveld analysis of X-ray diffraction patterns of Ti_3_(Al_1−*x*_Si*_x_*)C_2_ bulk samples with *x* = 0, 0.33, 0.5, 0.67 and 1: *a*, *c*—hexagonal unit cell parameters, $z_{Ti}$, $z_{C}$—positional parameters for Ti (4f) and C (4f) sites, respectively.

*x* (Si at.)	*a* (Å)	*c* (Å)	$z_{Ti}$	$z_{C}$
0	3.0746(2)	18.5670(6)	0.1273(3)	0.5692(8)
1/3	3.0736(1)	18.1500(3)	0.1310(2)	0.5720(5)
1/2	3.0718(1)	17.9511(3)	0.1326(2)	0.5711(5)
2/3	3.0713(1)	17.8588(2)	0.1334(1)	0.5723(4)
1	3.0674(2)	17.6723(5)	0.1353(2)	0.5743(9)

**Table 3 materials-14-03222-t003:** The approximated Debye temperatures ($\Theta_{D}^{l}$) and Sommerfeld coefficients (γ) determined from linear fits of the low temperature $C_{p}/T$ vs. $T^{2}$ plots (*T* < 7K) for the Ti_3_(Al_1−*x*_Si*_x_*)C_2_ compounds.

Parameter	Ti_3_AlC_2_	Ti_3_Al_2/3_Si_1/3_C_2_	Ti_3_Al_1/2_Si_1/2_C_2_	Ti_3_Al_1/3_Si_2/3_C_2_
$\Theta_{D}^{l}$(K)	484.2 ± 1.6	412.1 ± 0.4	472.0 ± 0.7	420.6 ± 0.5
γ(mJ⋅mol^−1^⋅K^−2^)	4.848 ± 0.006	4.653 ± 0.005	4.891 ± 0.003	5.893 ± 0.005

**Table 4 materials-14-03222-t004:** The Debye ($\Theta_{D}$), Einstein ($\Theta_{E_{i}}$) temperatures, and anharmonic coefficients (α) obtained from the fitting of Equation (4) to the $C_{p}$ vs. T dependences from Figure 7 for Ti_3_AlC_2_ and Ti_3_Al_1/2_Si_1/2_C_2_ compounds. In the refinement, the Sommerfeld (γ) coefficients were fixed at corresponding values from Table 3 and the multiplicities $m_{i}$ of the $\Theta_{E_{i}}$ parameters were set equal to 3, except of the $m_{3}$ (for $\Theta_{E_{3}}$) which was set equal to 6.

Parameter	Ti_3_AlC_2_	Ti_3_Al_1/2_Si_1/2_C_2_
$\Theta_{D}$ (K)	396.2 ± 4.1	371.9 ± 2.2
$\Theta_{E_{1}}$ (K)	299.6 ± 3.4	296.2 ± 2.1
$\Theta_{E_{2}}$ (K)	459.2 ± 4.8	506.2 ± 6.2
$\Theta_{E_{3}}$ (K)	652.9 ± 3.4	638.3 ± 4.3
$\Theta_{E_{4}}$ (K)	1107 ± 22	1098 ± 20
α(10^−5^⋅K^−1^)	9.1 ± 1.0	8.5 ± 0.5

**Table 5 materials-14-03222-t005:** Crystallographic information on the Ti_3_AlC_2_, Ti_3_SiC_2_, and Ti_3_Al_1−x_Si_x_C_2_ compounds.

Compound	Lattice Parameters (Å)	Atoms	Wyckoff Positions	x	y	z
Ti_3_AlC_2_	*a* = 3.0547 *b* = 3.0547 *c* = 18.453	Ti-1	2a	0	0	0
		Ti-2	4f	1/3	2/3	0.128
		Al	2b	0	0	0.25
		C	4f	1/3	2/3	0.064
Ti_3_Al_3/4_Si_1/4_C_2_	*a* = 6.090 *b* = 6.090 *c* = 18.396	Ti-1	6g	1/2	1/2	1/2
		Ti-2	2a	0	0	1/2
		Ti-3	12k	0.331	0.166	0.129
		Ti-4	4f	1/3	2/3	0.127
		Si/Al	2b	1/2	0	1/4
		Al/Si	6h	0.498	0.502	1/4
		C-1	12k	0.167	0.834	0.070
		C-2	4f	1/3	2/3	0.57
Ti_3_Al_1/2_Si_1/2_C_2_	*a* = 6.103 *b* = 6.103 *c* = 18.064	Ti-1	6n	0.5006	0.4999	0.246
		Ti-2	2g	0	0	0.247
		Ti-3	6n	0.167	0.833	0.117
		Ti-4	6n	0.834	0.166	0.378
		Ti-5	2h	1/3	2/3	0.375
		Ti-6	2i	2/3	1/3	0.113
		Si-1	3j	0.490	0.510	0
		Si-2	1b	0	0	1/2
		Al.-1	1a	0	0	0
		Al.-2	3k	0.493	0.507	1/2
		C-1	6n	0.166	0.834	0.317
		C-2	6n	0.833	0.167	0.176
		C-3	2h	1/3	2/3	0.177
		C-4	2i	2/3	1/3	0.318
Ti_3_Al_1/4_Si_3/4_C_2_	*a* = 6.157 *b* = 6.157 *c* = 18.012	Ti-1	6g	1/2	1/2	1/2
		Ti-2	2a	0	0	1/2
		Ti-3	12k	0.335	0.168	0.132
		Ti-4	4f	1/3	2/3	0.134
		Al	2b	0	0	1/4
		Si	6h	0.504	0.496	1/4
		C-1	12k	0.167	0.833	0.071
		C-2	4f	1/3	2/3	0.571
Ti_3_SiC_2_	*a* = 3.0422 *b* = 3.0422 *c* = 17.620	Ti-1	2a	0	0	0
		Ti-2	4f	1/3	2/3	0.135
		Si	2b	0	0	0.25
		C	4f	1/3	2/3	0

## Data Availability

The data presented in this study are available on request from the corresponding author.

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
