# Peer review of "Structure, Morphology, Heat Capacity, and Electrical Transport Properties of Ti3(Al,Si)C2 Materials"

_materials, 2021, doi:10.3390/ma14123222_

Round 1

Reviewer 1 Report

The manuscript by Goc et al. presents a thorough study of some pure MAX phases and solid solutions. Various experimental techniques are employed for their characterization (X-ray diffraction, scanning electron microscopy, heat capacity and electrical resistivity and magnetoresistance measurements). To complement the experimental characterization of these compounds, first-principles calculations have been carried out, and conclusions are drawn with regards to the high resistivity in connection with the density of states.
The manuscript is mostly well written and well presented, with the right amount of details. It would be interesting to learn more in the introduction about how these materials can be used in applications. 

A few remarks/corrections: 

Line 308: Why were the lattice parameters obtained from [18] rather than this work? All the relaxations could be done starting from the experimental values measured. 
Moreover, I suppose that the contribution of different phases were not included in the model? It seems they are quite large, from the XRD patterns analysis. Do the authors expect this would have some impact in the calculation results?

On p. 11, calculations of Fermi surfaces are shown and discussed, before any discussion on performing the actual calculations (from the following page). This should be at least referred to in the text.

On p. 12, the implications of the results on the density of states at Ef are discussed BEFORE presenting of the results. This reads quite bad and should be changed. Moreover, to help the reader, it would be desirable to also show the projected density of states on s/p/d orbitals alongside the total density of states in a plot. 

Eq.1 should be accompanied by a reference.

Since Table 4 only shows the parameters for 2 of the 4 compounds, this could be specified in the text when describing it.

Table 2 is not referenced in the text

Line 204-206: The linear temperature dependence of the heat capacity at the low temperature region reveals a metallic character of the MAX phase compounds studied. -> I supposed the linear dependence of the Cp/T vs T^2 instead

Line 68: synthesis, characterization of -> synthesis AND characterization of
Caption Fig.2: missing 'difference' in last sentence.
Legend in Fig.8: lines and colors not visible
Line 353: form -> from

Author Response

Line 308: Why were the lattice parameters obtained from [18] rather than this work? All the relaxations could be done starting from the experimental values measured. Moreover, I suppose that the contribution of different phases were not included in the model? It seems they are quite large, from the XRD patterns analysis. Do the authors expect this would have some impact in the calculation results?

>>> The contribution of other phases will be hard to calculate because the elementary cell need to be expanded, which increases drastically calculation time. We decided to calculate exemplary DOS of the pure, idealized, MAX phase only. Contribution or impurities probably will be negligible, because they form well crystalized, separate phases.

On p. 11, calculations of Fermi surfaces are shown and discussed, before any discussion on performing the actual calculations (from the following page). This should be at least referred to in the text.

>>> A proper reference to the next chapter was added (Line 320-323).

On p. 12, the implications of the results on the density of states at Ef are discussed BEFORE presenting of the results. This reads quite bad and should be changed. Moreover, to help the  reader, it would be desirable to also show the projected density of states on s/p/d orbitals alongside the total density of states in a plot.

>>> We added new plots showing total DOS, as well as  DOS for different orbitals. It is unusual to comment the results before presenting them, however it is only a short reference to the next chapter and explain to the reader what was the motivation of simulations.

Eq.1 should be accompanied by a reference.

>>> Proper reference was added.

Since Table 4 only shows the parameters for 2 of the 4 compounds, this could be specified in the text when describing it.

>>> We added description of data presented in the Table 4.

Table 2 is not referenced in the text.

>>> Table 2 is referenced in Line 147.

Line 204-206: The linear temperature dependence of the heat capacity at the low temperature region reveals a metallic character of the MAX phase compounds studied. -> I supposed the linear dependence of the Cp/T vs T^2 instead
Line 68: synthesis, characterization of -> synthesis AND characterization of Caption Fig.2: missing 'difference' in last sentence.
Legend in Fig.8: lines and colors not visible
Line 353: form -> from

>>> Corrections were made as suggested.

Reviewer 2 Report

report materials-1196261 

Title: Structure, Morphology, Heat Capacity and Electrical Transport Properties of the Ti3(Si,Al)C2 Materials  

Authors: Kamil Goc * , Janusz Przewoźnik , Katarzyna Chabior , Leszek Chlubny , Waldemar Tokarz , Tomasz Strączek , Jan Michalik , Jakub Jurczyk , Ivo Utke , Czesław Kapusta , Jerzy Lis  

Referal Round 01  

The authors study Si doped Ti3Al1-xSixC2 (x=0.33, 0.5, 0.67) MAX-phase compounds.  The materials are characterised with X-ray diffraction, heat capacity, electrical resistivity and magnetoresistance measurements. The authors claim that the results of resistivity do not depend in first instance on scattering of electrons due to defects. However the slightly lower magnitude of the magnetoresistance in this study compared to the one obtained for the undoped Ti 3AlC2 compound in previous studies, is attributed to a shorter mean free path of the electrical carriers, related to additional scattering due to a possible Ti/Si disorder.  

First, I suggest to explain all abreviations by first mentioning the term plain text and the abbreviation in brackets.  

Second the quality of the figures must be improved in the sense of rising the size of the symbols als rising all coat thicknesses.  

Third, the authors should commend on the issue, whether they included the interference effects for electronic transport and mean free paths in the sense of

D. Vollhardt and P. Wölfle, Phys. Rev. Lett. 45, 842 (1980).
D. Vollhardt and P. Wölfle, Phys. Rev. B 22, 4666 (1980).  

They should definitely broaden the research background towards including a comment to optical transport as in  

A. Lubatsch and R. Frank, PHYSICAL REVIEW RESEARCH 2, 013324 (2020) A. Lubatsch and R. Frank, Appl. Sci. 10, 1836 (2020)  

Fourth, the clarity of the abstract must be improved.  

Given those revisions done I might intend to recommend publication of this article.  

Author Response

1) Fig. 9 shows the Fermi surfaces. (a) The meaning of colors are not specified. What do the colors represent? A color map should be given. (b) The previous article by the authors, Ref. 17, also presents the Fermi surface of Ti3AlC2. This figure is significantly different from the one shown in Fig. 9. Why? Did the authors obtain these figures themselves? Then, they should have known the differences and should have commented on them. If taken from literature then, proper citation and again a discussion of the differences are necessary. (c) The electron energy bands for both compounds should be shown, as well. This way, the readers can have a better understanding of the presented Fermi surfaces.

>>> Plots showing band structure were added as suggested. The figure of the Fermi surface in the previous work has showed different band, that is why it is different from current picture. Different colour schemes were used only to increase band visibility.

2) DOS related parts: (a) First of all, the DOS of all relevant compounds should be plotted and the totally useless Table 6 should be discarded. Those numbers with a lot of zeros do not mean much and do not contribute to understanding. (b) The Sommerfeld coefficient \gamma is related to the DOS at the Fermi energy, N(E_F), through the simple free electron model. The values of the Sommerfeld coefficient obtained from the heat capacity measurements should be compared with the N(E_F) values obtained from the first-principles calculations. The necessary comparisons and interpretations should follow.

>>> A table with DOS was replaced with the additional figures (12&13) and the Sommerfeld coefficients obtained from calculations and experiment (line 285-295). We decided to present it in the heat capacity section and the reference to chapter about DOS was added.

3) Crystal structure data used in the first-principles calculations: Please state clearly if the lattice parameters and the free internal parameters, the z values associated with the 4f: 1/3 2/3 z positions are fully optimized or not for both end members. Likewise, clarify the situation for the two mixed ones, which are modeled on a 2 x 2 x 1 supercell. The c lattice parameters for the (1/3,2/3) and (2/3,1/3) compounds are identical, although the a lattice parameters are different. This is highly unlikely to happen. In addition, the values of the free atomic positions seem to be identical for these two compounds with different compositions (only a single set of values are listed in Table 5). In principle, the crystal structure data of the both compounds should be fully optimized separately. The compositions are different, so the free parameters should also be different.

>>>Lattice parameters are optimised to each compound separately. Table 5 was edited to make it more clear.

4) A figure showing the crystal structure of the 312 phase should be included in the manuscript.

>>>Figure showing 312 layered crystal structure (Figure 1) was added.

5) Line 163: Difference in volume is expressed as "0.16". What is that? There is no unit mentioned. If percentage, then should be stated clearly as so. In any case, I tried different ratios, but never got "0.16", I got essentially 1.4% as the deviation from the line.

>>> “0.16” is the different in the Si content (x) between the 0.5 and the value of x (~0.66) from theoretical dependence (blue) which poses the same volume value (~146,6 Å). We decided to edit this chapter and now the volumes are compared directly.

6) Lines 168-170: The change in lattice parameters c and a deserve some comments. a lattice parameter changes very little, c by a large amount. Is that due to difference in atomic radii? Or, perhaps, due to difference in the nature and/or strength of chemical bonding? Please, try to provide some explanation, why do you think you are getting such a result??

>>> It is an effect of anisotropic structure and substitution of Al atoms with smaller Si atoms. Proper explanation was added.

7) Typing errors and language related:

Lines 51-54: All capital letters appearing should be lower case.

Lines 84-85: Again, lower case: local density of states

Line 92: For #2: either a 2 on the product side or 1/2's on the reactants' side is necessary for a balanced equation.

Line 98: Define "HP".

Line 112: Lower case: density of states

Line 115: has been performed --> was performed

Line 244: of the of Debye --> of the Debye

Line 247: many Debye and Einstein --> many Einstein

Lines 305, 306: Lower case: density functional theory / generalized gradient approximation

Table 5, caption: 1-z and x as subscripts: ==> 1-x and x

>>>Typing errors were corrected.

Reviewer 3 Report

The manuscript presents the results of a combined experimental and computational study on Ti3(Si,Al)C2. These materials are referred as MAX phases, and they have been of intense interest in recent years. In this sense this manuscript is timely and can attract attention from the scientific community. The experimental aspects of the study seem to be fine, however, the parts on the first-principles study are highly inadequate. The presentation and discussion of the computational parts should be improved significantly. Therefore, I recommend major revision.

List of issues:

1) Fig. 9 shows the Fermi surfaces. (a) The meaning of colors are not specified. What do the colors represent? A color map should be given. (b) The previous article by the authors, Ref. 17, also presents the Fermi surface of Ti3AlC2. This figure is significantly different from the one shown in Fig. 9. Why? Did the authors obtain these figures themselves? Then, they should have known the differences and should have commented on them. If taken from literature then, proper citation and again a discussion of the differences are necessary. (c) The electron energy bands for both compounds should be shown, as well. This way, the readers can have a better understanding of the presented Fermi surfaces.

2) DOS related parts: (a) First of all, the DOS of all relevant compounds should be plotted and the totally useless Table 6 should be discarded. Those numbers with a lot of zeros do not mean much and do not contribute to understanding. (b) The Sommerfeld coefficient \gamma is related to the DOS at the Fermi energy, N(E_F), through the simple free electron model. The values of the Sommerfeld coefficient obtained from the heat capacity measurements should be compared with the N(E_F) values obtained from the first-principles calculations. The necessary comparisons and interpretations should follow.

3) Crystal structure data used in the first-principles calculations: Please state clearly if the lattice parameters and the free internal parameters, the z values associated with the 4f: 1/3 2/3 z positions are fully optimized or not for both end members. Likewise, clarify the situation for the two mixed ones, which are modeled on a 2 x 2 x 1 supercell. The c lattice parameters for the (1/3,2/3) and (2/3,1/3) compounds are identical, although the a lattice parameters are different. This is highly unlikely to happen. In addition, the values of the free atomic positions seem to be identical for these two compounds with different compositions (only a single set of values are listed in Table 5). In principle, the crystal structure data of the both compounds should be fully optimized separately. The compositions are different, so the free parameters should also be different.

4) A figure showing the crystal structure of the 312 phase should be included in the manuscript.

5) Line 163: Difference in volume is expressed as "0.16". What is that? There is no unit mentioned. If percentage, then should be stated clearly as so. In any case, I tried different ratios, but never got "0.16", I got essentially 1.4% as the deviation from the line.

6) Lines 168-170: The change in lattice parameters c and a deserve some comments. a lattice parameter changes very little, c by a large amount. Is that due to difference in atomic radii? Or, perhaps, due to difference in the nature and/or strength of chemical bonding? Please, try to provide some explanation, why do you think you are getting such a result??

7) Typing errors and language related:

Lines 51-54: All capital letters appearing should be lower case.

Lines 84-85: Again, lower case: local density of states

Line 92: For #2: either a 2 on the product side or 1/2's on the reactants' side is necessary for a balanced equation.

Line 98: Define "HP".

Line 112: Lower case: density of states

Line 115: has been performed --> was performed

Line 244: of the of Debye --> of the Debye

Line 247: many Debye and Einstein --> many Einstein

Lines 305, 306: Lower case: density functional theory / generalized gradient approximation

Table 5, caption: 1-z and x as subscripts: ==> 1-x and x

Author Response

First, I suggest to explain all abreviations by first mentioning the term plain text and the abbreviation in brackets.  

Second the quality of the figures must be improved in the sense of rising the size of the symbols als rising all coat thicknesses.  

>>> All abbreviations were preceded by the proper explanation and the fonts and line thickness were increased.

Third, the authors should commend on the issue, whether they included the interference effects for electronic transport and mean free paths in the sense of  

  1. Vollhardt and P. Wölfle, Phys. Rev. Lett. 45, 842 (1980).
    D. Vollhardt and P. Wölfle, Phys. Rev. B 22, 4666 (1980).  

They should definitely broaden the research background towards including a comment to optical transport as in  

  1. Lubatsch and R. Frank, PHYSICAL REVIEW RESEARCH 2, 013324 (2020) A. Lubatsch and R. Frank, Appl. Sci. 10, 1836 (2020)  

>>> We added the comment and reference to the Anderson localisation effects (Line 380-386).

Fourth, the clarity of the abstract must be improved.  

>>>Abstract was reduced and edited in order to improve its clarity.

Round 2

Reviewer 2 Report

materials-1196261 

Title: Structure, Morphology, Heat Capacity and Electrical Transport Properties of the Ti3(Si,Al)C2 Materials   Authors Kamil Goc * , Janusz Przewoźnik , Katarzyna Chabior , Leszek Chlubny , Waldemar Tokarz , Tomasz Strączek , Jan Michalik , Jakub Jurczyk , Ivo Utke , Czesław Kapusta , Jerzy Lis     While I can see that the manuscript has been improved and the authors were willing to include some of my comments to their manuscript, I am missing the answers to my report in a sense of the authors response, and I can only see the answers to the second report.   I do not understand why the authors believe that Anderson localization should be restricted to verification via optical measurements, maybe they can add another clarifying sentence.    

Author Response

REVIEWER 2.

While I can see that the manuscript has been improved and the authors were willing to include some of my comments to their manuscript, I am missing the answers to my report in a sense of the authors response, and I can only see the answers to the second report.   I do not understand why the authors believe that Anderson localization should be restricted to verification via optical measurements, maybe they can add another clarifying sentence.  

>>>Restricting verification of Anderson localization to optical measurements was indeed not justified, so this part of the sentence was removed.

Here we present answers for your comments from the first round of revisions:

First, I suggest to explain all abreviations by first mentioning the term plain text and the abbreviation in brackets.  

Second the quality of the figures must be improved in the sense of rising the size of the symbols als rising all coat thicknesses.  

>>> After corrections all abbreviations are preceded by the proper explanation and the fonts and line thickness were increased.

Third, the authors should commend on the issue, whether they included the interference effects for electronic transport and mean free paths in the sense of  

  1. Vollhardt and P. Wölfle, Phys. Rev. Lett. 45, 842 (1980).
    D. Vollhardt and P. Wölfle, Phys. Rev. B 22, 4666 (1980).  

They should definitely broaden the research background towards including a comment to optical transport as in  

  1. Lubatsch and R. Frank, PHYSICAL REVIEW RESEARCH 2, 013324 (2020) A. Lubatsch and R. Frank, Appl. Sci. 10, 1836 (2020)  

>>> We added the comment and reference to the Anderson localisation effects (Line 380).

Fourth, the clarity of the abstract must be improved.  

>>>Abstract was reduced and re-edited in order to improve its clarity.

Reviewer 3 Report

The authors revised the manuscript according to the remarks made in the first review round. But, these modifications made some serious mistakes more obvious. Basically, the first-principles calculations related parts have to be written afresh, they are not valid as they stand. The authors failed to realize some simple but fatal mistakes in their computational work. The language also has become more problematic. I recommend major and very careful revision subject to subsequent review.

Scientific issues:

1) Calculations related to Table 5: (a) First, let's assume that the results presented in this table are valid (they are not). The a lattice parameter decreases with x, Si content. This is fine. But, the optimized c lattice parameters for x = 1/3 and 2/3 are the same! This cannot be. (b) Actually, the authors performed calculations for x = 1/4 and x = 3/4, but they have still not realized what they had done! I wrote in the first round that the super cell used is a 2 x 2 x 1 one, meaning 4 times larger. The Al/Si disorder takes place in 2b position. 4 times larger super cell gives 8 sites, and you cannot partition 8 sites between two elements with a ratio of 2-to-1! None of the 11 authors were able to notice that 8 sites being split as 6h + 2b yields a ratio of 3-to-1, not 2-to-1. Therefore, comparing Table 5-based results with the experimental results is meaningless. Experiment has 1/3,2/3 and 2/3,1/3, but calculations are for 1/4,3/4 and 3/4,1/4 compositions. The authors should redo all these calculations for the proper (or, "correct") compositions. They can use a super cell of 1 x 1 x 3 in the same space group, that would be the simplest case. (c) Moreover, why omit the 1/2,1/2 composition? This is also an experimental point, and to my eyes, it is an important case. Fig. 4 shows clearly that: (i) V(x) seems to have two linear regions with different slopes, and x = 0.5 is the border; (ii) Same for c(x); (iii) a(x=0.5) has a smaller value than the line joining 1/3 and 2/3 cases. This suggests that something is special with x = 0.5. To understand it, one should perform a calculation for that composition. Simulating x=0.5 is much simpler. Use the subgroup no. 187 with identity matrix as the transformation matrix and the shift vector of P = [0 0 1/4]. This transformation splits the 2b position into 2 sites in SG. 187: 0 0 0 and 0 0 1/2. Assign Si to one, Al to other, and there you go: you have Ti3(Al_0.5Si_0.5)C2. They can keep the present results, as well (after resolving the c lattice parameter issue), and at the end they will have many points between x = 0 to x = 1. They can plot a versus x, c versus x, etc. These can be compared with their experimental counterparts (Fig. 4). It would be interesting to see if x=0.5 emerges as a special case in calculations, too. Experimentally you have only a handful of points, this is why first-principles calculations are useful, you can extend the data base and reach to a deeper understanding level. The manuscript is currently very weak in this regard. (d) Another idea is to use the virtual crystal approximation (VCA). Since Si and Al are neighbors in the Periodic Table, this approach is expected to work quite well. Place a virtual atom of Z(x) = 13 + x, x between 0 and 1 at the 2b position. This way one can simulate any composition. It may be interesting to compare some VCA results with those of the corresponding ordered super cell calculations. However, this is just an idea for the authors to keep in mind for other possible similar projects. They do not have to perform such calculations for this manuscript.

2) The paragraph between lines 293 and 297: There are two problems here. One of them is comparing the value of DOS at Fermi energy (EF), N(EF), for a compound to the sum of N(EF)'s of the elements forming it. DOS and the electron energy band dispersions reflect the hybridization of atomic orbitals (chemical bonding) in that compound. Different compounds have different atomic interactions due to different crystal structures and stoichiometry, and so they will have different bands, and different DOS, thus such comparisons are not valid. The other problem is that the authors seem to imply that small value of N(EF) corresponds to a smaller electron density at the Fermi level. These are two different quantities. DOS and N(EF) are concepts in Hilbert space. They are about the distributions of energy eigenvalues over the energy. Electron density, on the other hand, is a quantity in real or position space with a unit of charge per volume. This misleading paragraph should be removed.

3) Fig. 8 and related text: (a) The figure shows a value for Sommerfeld coefficient for the x = 1 case, Ti3SiC2. However, in the manuscript there is no mention of physical property measurements being carried out for this compound. So, where did you get this value from? If from literature, then the source should be cited. If own result, then the manuscript should state this clearly. (b) The text about this figure claims that experimental values are smaller than theoretical ones. But the figure shows just the opposite. Such trivial mistakes imply carelessness on authors' part. Being fast is good, but if you make many mistakes as a result of speed, then it will take much more time to reach your goal. Please, take your time, think carefully, and only then write.

4) Line 162: What does "meaningful disorder" mean? The statement that XRD patterns reveal no evidence for "meaningful disorder" is very very confusing, because after all these Ti3Al_(1-x)Si_xC2 compounds are expected to have Al/Si disorder, isn't it? If the XRD did not show "disorder", then which compound did you measure? Since it is not mentioned in the text, I assume that there are no indications of a superstructure in these XRD patterns which might imply an ordering of Si and Al atoms in some super cell. Meaning, Al and Si are truly disordered, so, how should we understand this sentence?

Language related:

1) Abstract: First of all, the compounds with nonzero x are solid solutions or alloys, and it seems like it is possible to go from Ti3AlC2 to Ti3SiC2 (x=0 to x =1) in an essentially continuous manner. Therefore, one can equally well say that we are doping Si into the former, or doping Al into the latter. I see no reason to choose one over the other. Consequently, all phrases of "Si doping" in the whole manuscript should be removed and replaced with a more correct phrase. Corrections in abstract:

A study of Ti3Al_(1-x)Si_xC2 (..) MAX-phase alloys

The results show ... and metallic properties with high residual resistivities.

The resistivity weakly varies with Si content and shows ...

The Debye ... show slight variations with Si content, with a decreasing ... an increasing one for the latter.

Experimental results were supported by band structure calculations whose results are consistent with the specific heat, ... magnetoresistance measurements.

Line 73: comparing --> compared

Line 78: Following our ... on Ti3AlC2, ...

Line 163: ... 10% in the Si doped samples --> 10% in the samples of solid solution materials

Figure 4: This figure needs the labels a) and b).

Line 232: divergences --> differences

Line 264: plus --> and

Figure 11: The caption should mention the band dispersion figures, as well.

Author Response

REVIEWER 3.

The authors revised the manuscript according to the remarks made in the first review round. But, these modifications made some serious mistakes more obvious. Basically, the first-principles calculations related parts have to be written afresh, they are not valid as they stand. The authors failed to realize some simple but fatal mistakes in their computational work. The language also has become more problematic. I recommend major and very careful revision subject to subsequent review.

Scientific issues:

1) Calculations related to Table 5: (a) First, let's assume that the results presented in this table are valid (they are not). The a lattice parameter decreases with x, Si content. This is fine. But, the optimized c lattice parameters for x = 1/3 and 2/3 are the same! This cannot be. (b) Actually, the authors performed calculations for x = 1/4 and x = 3/4, but they have still not realized what they had done! I wrote in the first round that the super cell used is a 2 x 2 x 1 one, meaning 4 times larger. The Al/Si disorder takes place in 2b position. 4 times larger super cell gives 8 sites, and you cannot partition 8 sites between two elements with a ratio of 2-to-1! None of the 11 authors were able to notice that 8 sites being split as 6h + 2b yields a ratio of 3-to-1, not 2-to-1. Therefore, comparing Table 5-based results with the experimental results is meaningless. Experiment has 1/3,2/3 and 2/3,1/3, but calculations are for 1/4,3/4 and 3/4,1/4 compositions. The authors should redo all these calculations for the proper (or, "correct") compositions. They can use a super cell of 1 x 1 x 3 in the same space group, that would be the simplest case. (c) Moreover, why omit the 1/2,1/2 composition? This is also an experimental point, and to my eyes, it is an important case. Fig. 4 shows clearly that: (i) V(x) seems to have two linear regions with different slopes, and x = 0.5 is the border; (ii) Same for c(x); (iii) a(x=0.5) has a smaller value than the line joining 1/3 and 2/3 cases. This suggests that something is special with x = 0.5. To understand it, one should perform a calculation for that composition. Simulating x=0.5 is much simpler. Use the subgroup no. 187 with identity matrix as the transformation matrix and the shift vector of P = [0 0 1/4]. This transformation splits the 2b position into 2 sites in SG. 187: 0 0 0 and 0 0 1/2. Assign Si to one, Al to other, and there you go: you have Ti3(Al_0.5Si_0.5)C2. They can keep the present results, as well (after resolving the c lattice parameter issue), and at the end they will have many points between x = 0 to x = 1. They can plot a versus x, c versus x, etc. These can be compared with their experimental counterparts (Fig. 4). It would be interesting to see if x=0.5 emerges as a special case in calculations, too. Experimentally you have only a handful of points, this is why first-principles calculations are useful, you can extend the data base and reach to a deeper understanding level. The manuscript is currently very weak in this regard. (d) Another idea is to use the virtual crystal approximation (VCA). Since Si and Al are neighbors in the Periodic Table, this approach is expected to work quite well. Place a virtual atom of Z(x) = 13 + x, x between 0 and 1 at the 2b position. This way one can simulate any composition. It may be interesting to compare some VCA results with those of the corresponding ordered super cell calculations. However, this is just an idea for the authors to keep in mind for other possible similar projects. They do not have to perform such calculations for this manuscript.

>>> We are grateful that you found our mistake in attributing our calculations to x =1/3 and 2/3 and not to 1/4 and 3/4 as it correctly should be. We also appreciate your comments and suggestions on the theoretical calculations.  The plot of  values (Fig. 8) was corrected accordingly and we were able to perform calculations for the x = 1/2 composition according to your suggestion and added the data obtained to the manuscript.  We also replaced plots of DOS for x = 1/4 and 3/4 with DOS for x = 1/2.

2) The paragraph between lines 293 and 297: There are two problems here. One of them is comparing the value of DOS at Fermi energy (EF), N(EF), for a compound to the sum of N(EF)'s of the elements forming it. DOS and the electron energy band dispersions reflect the hybridization of atomic orbitals (chemical bonding) in that compound. Different compounds have different atomic interactions due to different crystal structures and stoichiometry, and so they will have different bands, and different DOS, thus such comparisons are not valid. The other problem is that the authors seem to imply that small value of N(EF) corresponds to a smaller electron density at the Fermi level. These are two different quantities. DOS and N(EF) are concepts in Hilbert space. They are about the distributions of energy eigenvalues over the energy. Electron density, on the other hand, is a quantity in real or position space with a unit of charge per volume. This misleading paragraph should be removed.

>>> This misleading paragraph was removed.

3) Fig. 8 and related text: (a) The figure shows a value for Sommerfeld coefficient for the x = 1 case, Ti3SiC2. However, in the manuscript there is no mention of physical property measurements being carried out for this compound. So, where did you get this value from? If from literature, then the source should be cited. If own result, then the manuscript should state this clearly. (b) The text about this figure claims that experimental values are smaller than theoretical ones. But the figure shows just the opposite. Such trivial mistakes imply carelessness on authors' part. Being fast is good, but if you make many mistakes as a result of speed, then it will take much more time to reach your goal. Please, take your time, think carefully, and only then write.

>>> This value was obtained from the experiment and was left by mistake from the earlier version of the manuscript. As the Ti3SiC2 sample contains too much impurities we decided not to show the experimental data for it.   In the present, revised version it is removed from the plot.

4) Line 162: What does "meaningful disorder" mean? The statement that XRD patterns reveal no evidence for "meaningful disorder" is very very confusing, because after all these Ti3Al_(1-x)Si_xC2 compounds are expected to have Al/Si disorder, isn't it? If the XRD did not show "disorder", then which compound did you measure? Since it is not mentioned in the text, I assume that there are no indications of a superstructure in these XRD patterns which might imply an ordering of Si and Al atoms in some super cell. Meaning, Al and Si are truly disordered, so, how should we understand this sentence?

>>> The statement on the lack of evidence of a meaningful disorder was not justified in view of Si/Al intermixing and we removed it.

Language related:

1) Abstract: First of all, the compounds with nonzero x are solid solutions or alloys, and it seems like it is possible to go from Ti3AlC2 to Ti3SiC2 (x=0 to x =1) in an essentially continuous manner. Therefore, one can equally well say that we are doping Si into the former, or doping Al into the latter. I see no reason to choose one over the other. Consequently, all phrases of "Si doping" in the whole manuscript should be removed and replaced with a more correct phrase. Corrections in abstract:

A study of Ti3Al_(1-x)Si_xC2 (..) MAX-phase alloys

The results show ... and metallic properties with high residual resistivities.

The resistivity weakly varies with Si content and shows ...

The Debye ... show slight variations with Si content, with a decreasing ... an increasing one for the latter.

Experimental results were supported by band structure calculations whose results are consistent with the specific heat, ... magnetoresistance measurements.

Line 73: comparing --> compared

Line 78: Following our ... on Ti3AlC2, ...

Line 163: ... 10% in the Si doped samples --> 10% in the samples of solid solution materials

Figure 4: This figure needs the labels a) and b).

Line 232: divergences --> differences

Line 264: plus --> and

Figure 11: The caption should mention the band dispersion figures, as well.

>>> All language related issues and problems have been corrected, as suggested.

Round 3

Reviewer 3 Report

The manuscript is in a better shape, but there are still serious issues. The authors do not try to make sense of the results they obtain by computation. Also, they do not check the numbers listed in the tables properly (or they fill these tables in a careless manner). So, once again, I recommend major revision due to the reasons provided below.

Scientific issues:

1) Table 2: The c lattice parameters for x = 1/2 and 2/3 are listed as 17.8588 AA. This information contradicts the graph of Fig. 4(b). The lattice parameter c is expected to decrease with x. Please correct.

2) The theoretically obtained lattice parameters (Table 5): The authors should have checked how meaningful their results were. I summarize their results for the lattice parameter a as a function x below:

0.0   3.0547 AA

1/4  3.0450 AA

1/2  3.0515 AA

3/4  3.0785 AA

1.0  3.0422 AA

As can be seen very clearly, the variation of a with x does not agree with the experimental findings. The lattice parameter a should decrease with increasing x (ie. increasing Si content), but here we observe a zig-zag behavior. Why is it that the authors do not mention this, do not comment on this? That the authors ignore this clear discrepancy is a serious problem. The authors should check their calculations (convergence criteria, sufficiency of various computational parameters, etc.), and should explain why they obtain this type of variation. They should also look at the variation of unit cell volume with x. Whatever you have in experiment and that can be calculated, then you should calculate that quantity and compare it directly with its experimental counterpart.

Other issues:

Lines 333-338: This part should be written completely anew. It provides confusing information as it stands. For ex., Table 5 lists the basic crystallographic data for x = 0, ..., 1 obtained as a result of optimization calculations, but the authors write "Table 5 presents the Ti3AlC2 structural data ...". The authors write "These parameters were obtained from [20] after minimization ...". This is confusing, because [20] must be experimental data, so how should the reader understand this sentence? It should be written with more care. The phrase "Some ... positions with free parameters ..." is also confusing, because in principle, one should optimize all free positional parameters. What does "Some" mean in this context? In addition, the ordered super cells for x = 1/4, 1/2 and 3/4 should all be described clearly, like 2 x 2 x 1 super cell in space group 194, etc.

Line 19: resistivity of --> resistivity

Line 25: band structure --> band structure calculations

Line 390: Hot Pressing --> lower case

Lines 391, 392: capital letters --> lower case

Line 395: upon Si doping --> with Si content

Author Response

Comments and Suggestions for Authors
The manuscript is in a better shape, but there are still serious issues. The authors do not try to make sense of the results they obtain by computation. Also, they do not check the numbers listed in the tables properly (or they fill these tables in a careless manner). So, once again, I recommend major revision due to the reasons provided below.
Scientific issues:
1) Table 2: The c lattice parameters for x = 1/2 and 2/3 are listed as 17.8588 AA. This information contradicts the graph of Fig. 4(b). The lattice parameter c is expected to decrease with x. Please correct.
>>> The error was corrected.

2) The theoretically obtained lattice parameters (Table 5): The authors should have checked how meaningful their results were. I summarize their results for the lattice parameter a as a function x below:
0.0   3.0547 AA
1/4  3.0450 AA
1/2  3.0515 AA
3/4  3.0785 AA
1.0  3.0422 AA
As can be seen very clearly, the variation of a with x does not agree with the experimental findings. The lattice parameter a should decrease with increasing x (ie. increasing Si content), but here we observe a zig-zag behavior. Why is it that the authors do not mention this, do not comment on this? That the authors ignore this clear discrepancy is a serious problem. The authors should check their calculations (convergence criteria, sufficiency of various computational parameters, etc.), and should explain why they obtain this type of variation. They should also look at the variation of unit cell volume with x. Whatever you have in experiment and that can be calculated, then you should calculate that quantity and compare it directly with its experimental counterpart.

>>> The calculated in DFT-LAPW methods lattice parameters accuracy in relation to experimental one depends on a system [http://wien2k.at/events/ws2019/Schwarz_DFT-LAPW.pdf , page 37.]. For example, we also calculated optimised V and c/a for the Ti3Al1/2Si1/2C2 in the structure 187 with Al and Si in separate layers and we got a = 3.0871 Å, but if we mix Si an Al within layers into the 2x2x1 superstructure this parameter is lower and amounts to 3.0517 Å. Thus if the Al/Si occupation in the Ti3Al1/4Si3/4C2 were fully random, which we cannot resolve, we would probably get a lower “a” parameter value. However, this zig-zag dependency needs further study, which cannot be done in the scope of this paper.

In our cases the volume and “c/a” optimizations were performed with the same criteria and way for all the samples. From the point of view of energy minima we got sufficient number of points, all laying on fitted curve to get good fit to estimate optimal “V” and “c/a” value. All free parameters in site positions were adjusted according to force minimization.
Other issues:
Lines 333-338: This part should be written completely anew. It provides confusing information as it stands. For ex., Table 5 lists the basic crystallographic data for x = 0, ..., 1 obtained as a result of optimization calculations, but the authors write "Table 5 presents the Ti3AlC2 structural data ...". The authors write "These parameters were obtained from [20] after minimization ...". This is confusing, because [20] must be experimental data, so how should the reader understand this sentence? It should be written with more care. The phrase "Some ... positions with free parameters ..." is also confusing, because in principle, one should optimize all free positional parameters. What does "Some" mean in this context? In addition, the ordered super cells for x = 1/4, 1/2 and 3/4 should all be described clearly, like 2 x 2 x 1 super cell in space group 194, etc.

>>> The respective corrections were introduced to the text.

Line 19: resistivity of --> resistivity
Line 25: band structure --> band structure calculations
Line 390: Hot Pressing --> lower case
Lines 391, 392: capital letters --> lower case
Line 395: upon Si doping --> with Si content

>>> Corrections were done according to the suggestions.